# HGMD: Rethinking Hard Sample Distillation for GNN-to-MLP Knowledge Distillation

## Abstract

To bridge the gaps between powerful Graph Neural Networks (GNNs) and lightweight Multi-Layer Perceptron (MLPs), GNN-to-MLP Knowledge Distillation (KD) proposes to distill knowledge from a well-trained teacher GNN into a student MLP. A counter-intuitive observation is that *"better teacher, better student"* does not always hold true for GNN-to-MLP KD, which inspires us to explore what are the criteria for better GNN knowledge samples (nodes). In this paper, we revisit the knowledge samples in teacher GNNs from the perspective of **hardness** rather than **correctness**, and identify that hard sample distillation may be a major performance bottleneck of existing KD algorithms. The GNN-to-MLP KD involves two different types of hardness, one student-free *knowledge hardness* describing the inherent complexity of GNN knowledge, and the other student-dependent *distillation hardness* describing the difficulty of teacher-to-student distillation. In this paper, we propose a novel *Hardness-aware GNN-to-MLP Distillation* (HGMD) framework, which models both knowledge and distillation hardness and then extracts a hardness-aware subgraph for each sample separately, where a harder sample will be assigned a larger subgraph. Finally, two hardness-aware distillation schemes (i.e., HGMD-weight and HGMD-mixup) are devised to distill subgraph-level knowledge from teacher GNNs into the corresponding nodes of student MLPs. As non-parametric distillation, HGMD does not involve any additional learnable parameters beyond the student MLPs, but it still outperforms most of the state-of-the-art competitors. For example, HGMD-mixup improves over the vanilla MLPs by 12.95% and outperforms its teacher GNNs by 2.48% averaged over seven real-world datasets and three GNN architectures.

## 1 Introduction

Recently, the emerging Graph Neural Networks (GNNs) (Wu et al., 2020; Zhou et al., 2020) have demonstrated their powerful capability in handling various graph-structured data. Benefiting from the powerful topology awareness enabled by message passing, GNNs have achieved great academic success. However, the neighborhood-fetching latency arising from data dependency in GNNs makes it still less popular for practical deployment, especially in computational-constraint applications. In contrast, Multi-Layer Perceptrons (MLPs) are free from data dependencies among neighboring nodes and infer much faster than GNNs, but at the cost of suboptimal performance. To bridge these two worlds, GLNN (Zhang et al., 2022) proposes GNN-to-MLP Knowledge Distillation (KD), which extracts informative knowledge from a teacher GNN and then injects it into a student MLP.

A long-standing intuitive idea about knowledge distillation is *"better teacher, better student"*. In other words, distillation from a better teacher is expected to yield a better student, since a better teacher can usually capture more informative knowledge from which the student can benefit. However, some recent work has challenged this intuition, arguing that it does not hold true in all cases, i.e., distillation from a larger teacher, typically with more parameters and high accuracy, may be inferior to distillation from a smaller, less accurate teacher (Mirzadeh et al., 2020; Shen et al., 2021; Stanton et al., 2021; Zhu et al., 2022). To illustrate this, we show the rankings of three teacher GNNs, including Graph Convolutional Network (GCN) (Kipf & Welling, 2016), Graph Attention Network (GAT) (Veličković et al., 2017), and GraphSAGE (Hamilton et al., 2017), on seven datasets, as well as their corresponding distilled MLPs in Fig. 1(a), from which we observe that GCN is the best teacher on the *Arxiv* dataset, but its distilled student MLP performs the poorest. There have been many previous works (Jafari et al., 2021; Zhu & Wang, 2021; Qiu et al., 2022) delving into

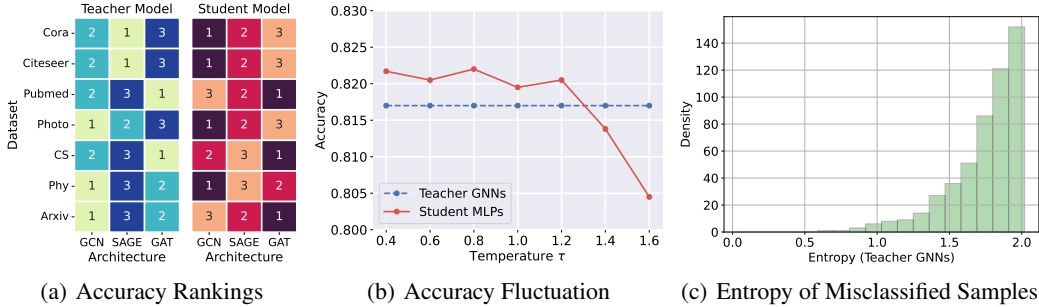

| (a) Accuracy Rankings | (b) Accuracy Fluctuation | (c) Entropy of Misclassified Samples |

Figure 1: *(a)* Accuracy rankings of three teacher GNNs (GCN, SAGE, and GAT) and their corresponding student MLPs on seven datasets. *(b)* Accuracy fluctuations of the teacher GCN and the student MLP w.r.t temperature coefficient $\tau$ on the Cora dataset. *(c)* Histogram of the information entropy of GNN knowledge for those samples misclassified by student MLPs on the Cora dataset.

this issue, but most of them attribute this counter-intuitive observation to the *capacity mismatch* between the teacher and student models. In other words, a student with fewer parameters may fail to "understand" the high-order semantic knowledge captured by a teacher with numerous parameters.

However, we found that the above popular explanation from a model capacity perspective may hold true for knowledge distillation in computer vision, but fails in the graph domain. For a teacher GCN and a student MLP with the same amount of parameters (i.e., the same layer depth and width), we plot the accuracy fluctuations of the teacher and distilled student with respect to the distillation temperature $\tau$ in Fig. 1(b). It can be seen that while the temperature $\tau$ does not affect the teacher's accuracy, it may influence the hardness of GNN knowledge, which in turn leads to different student's accuracy. This suggests that the model capability mismatch may not be sufficient to explain why "same (accuracy) teachers, different (accuracy) students" occurs during GNN-to-MLP distillation.

**Present Work.** In this paper, we rethink what exactly are the criteria for "better" knowledge samples (nodes) in teacher GNNs from the perspective of **hardness** rather than **correctness**, which has been rarely studied in previous works. The motivational experiment in Fig. 1(c) indicates that most GNN knowledge of samples misclassified by student MLPs is distributed in the high-entropy zones, which suggests that GNN knowledge samples with higher uncertainty are usually harder to be correctly distilled. Furthermore, we explore the roles played by GNN knowledge samples of different hardness during distillation and identify that *hard sample distillation may be a major performance bottleneck* of existing KD algorithms. As a result, to provide more supervision for the distillation of those hard samples, we propose a non-parametric Hardness-aware GNN-to-MLP Distillation (HGMD) framework. The proposed framework first models both knowledge and distillation hardness, then extracts a hardness-aware subgraph (the harder, the larger) for each sample separately, and finally applies two distillation schemes (i.e., HGMD-weight and HGMD-mixup) to distill the *subgraph-level knowledge* from teacher GNNs into the corresponding nodes of student MLPs.

Our main contributions are: (1) We are the first to identify that hard sample distillation is the main bottleneck that limits the performance of existing GNN-to-MLP KD algorithms, and more importantly, we have described in detail what it represents, what impact it has, and how to deal with it. (2) We decouple two different hardnesses, i.e., knowledge hardness and distillation hardness, and propose to distill knowledge in a hardness-aware manner to provide more supervision for hard samples. (3) We devise two distillation schemes for hard knowledge distillation. Despite not involving any additional parameters, they are still comparable to or even better than state-of-the-art competitors.

## 2 RELATED WORK

**GNN-to-GNN Knowledge Distillation.** Recent years have witnessed the great success of GNNs in handling graph-structured data. However, most existing GNNs share the de facto design that relies on message passing to aggregate features from neighborhoods, which may be one major source of latency in GNN inference. To address this problem, several previous works on graph distillation try to distill knowledge *from large teacher GNNs to smaller student GNNs*, termed as GNN-to-GNN knowledge distillation (KD) (Lassance et al., 2020; Zhang et al., 2020a; Ren et al., 2021; Joshi et al., 2021), including RDD (Zhang et al., 2020b), TinyGNN (Yan et al., 2020), LSP (Yang et al., 2020),

GraphAKD (He et al., 2022), GNN-SD (Chen et al., 2020b), and FreeKD (Feng et al., 2022), etc. However, both teacher and student models in the above works are GNNs, making these designs still suffer from the neighborhood-fetching latency arising from the data dependency in GNNs.

**GNN-to-MLP Knowledge Distillation.**   To bridge the gaps between powerful GNNs and lightweight MLPs, the other branch of graph knowledge distillation is to directly distill from teacher GNNs to lightweight student MLPs, termed GNN-to-MLP KD. For example, GLNN (Zhang et al., 2022) directly distills knowledge from teacher GNNs to vanilla MLPs by imposing KL-divergence between their logits. Instead, CPF (Yang et al., 2021) improves the performance of student MLPs by incorporating label propagation in MLPs, which may further burden the inference latency. Besides, FF-G2M (Wu et al., 2023a) propose to factorize GNN knowledge into low- and high-frequency components in the spectral domain and propose a novel framework to distill both low- and high-frequency knowledge from teacher GNNs into student MLPs. Moreover, RKD-MLP (Anonymous, 2023) takes the reliability of GNN knowledge into account and adopts a meta-policy to filter out unreliable GNN knowledge. Despite the great progress, most of these GNN-to-MLP KD methods have focused on how to make better use of those simple samples, while little effort has been made on those hard samples. However, we have found in this paper that hard sample distillation may be a main bottleneck that limits the performance of existing GNN-to-MLP KD algorithms.

## 3 METHODOLOGY

### 3.1 PRELIMINARIES AND NOTATIONS

Given a graph $\mathcal{G} = (\mathcal{V}, \mathcal{E})$, where $\mathcal{V} = \{v_1, v_2, \cdots, v_N\}$ and $\mathcal{E} \subseteq \mathcal{V} \times \mathcal{V}$ denote the node set and edge set, respectively. In addition, $\mathbf{X} \in \mathbb{R}^{N \times d}$ and $\mathbf{A} \in [0,1]^{N \times N}$ denotes the feature matrix and adjacency matrix, where each node $v_i \in \mathcal{V}$ is associated with a $d$-dimensional features vector $\mathbf{x}_i \in \mathbb{R}^d$ and $\mathbf{A}_{i,j} = 1$ iff $(v_i, v_j) \in \mathcal{E}$. Consider node classification in a transductive setting in which only a subset of node $\mathcal{V}_L \in \mathcal{V}$ with corresponding labels $\mathcal{Y}_L$ are known, we denote the labeled set as $\mathcal{D}_L = (\mathcal{V}_L, \mathcal{Y}_L)$ and unlabeled set as $\mathcal{D}_U = (\mathcal{V}_U, \mathcal{Y}_U)$, where $\mathcal{V}_U = \mathcal{V} \backslash \mathcal{V}_L$. The objective of GNN-to-MLP knowledge distillation is to first train a teacher GNN $\mathbf{Z} = f_\theta^{\mathcal{T}}(\mathbf{A}, \mathbf{X})$ on the labeled data $\mathcal{D}_L$, and then distill knowledge from the teacher GNN into a student MLP $\mathbf{H} = f_\gamma^{\mathcal{S}}(\mathbf{X})$ by imposing KL-divergence $\mathcal{D}_{KL}(\cdot, \cdot)$ between their label distributions on the node set $\mathcal{V}$, as follows

$$\mathcal{L}_{\text{KD}} = \frac{1}{|\mathcal{V}|} \sum_{i \in \mathcal{V}} \mathcal{D}_{KL}\Big(\sigma\left(\mathbf{z}_i/\tau\right), \sigma\left(\mathbf{h}_i/\tau\right)\Big), \tag{1}$$

where $\sigma(\cdot) = \text{softmax}(\cdot)$ is the activation function, and $\tau$ is the distillation temperature. Beisdes, $\mathbf{z}_i$ and $\mathbf{h}_i$ are the node embeddings of node $v_i$ in $\mathbf{Z}$ and $\mathbf{H}$, respectively. Once knowledge distillation is done, the distilled MLP can be used to infer the ground-truth label $y_i \in \mathcal{Y}_U$ for unlabeled data.

**Knowledge Hardness.**   Inspired by the experiment in Fig. 1(c), where GNN knowledge samples with higher entropy are harder to be correctly distilled into the student MLPs, we propose to use the information entropy $\mathcal{H}(\mathbf{z}_i)$ of node $v_i$ as a measure of its knowledge hardness, as follows

$$\mathcal{H}(\mathbf{z}_i) = -\sum_j \sigma\big(\mathbf{z}_{i,j}/\tau\big)\log\big(\sigma\left(\mathbf{z}_{i,j}/\tau\right)\big). \tag{2}$$

We default to using Eq. (2) for measuring the knowledge hardness in this paper and delay discussions on distillation hardness. See **Appendix F** for more results on other knowledge hardness metrics.

### 3.2 PERFORMANCE BOTTLENECK: HARD SAMPLE DISTILLATION

Recent years have witnessed the great success of knowledge distillation and a surge of related distillation techniques. As the research goes deeper, the rationality of *"better teacher, better student"* has been increasingly challenged. A lot of earlier works (Jafari et al., 2021; Son et al., 2021) have found that as the performance of the teacher model improves, the accuracy of the student model may unexpectedly gets worse. Most of the existing works attribute such counter-intuitive observation to the capacity mismatch between the teacher and student models. In other words, a smaller student may have difficulty "understanding" the high-order semantic knowledge captured by a large teacher. Although this problem has been well studied in computer vision, little work has been devoted to whether it exists in graph knowledge distillation, what it arises from, and how to deal with it. In

this paper, we get the same observation during GNN-to-MLP distillation that better teachers do not necessarily lead to better students in Fig. 1(a), but we find that this has little to do with the popular idea of capacity mismatch. This is because, unlike common visual backbones with very deep layers in computer vision, GNNs tend to suffer from the undesired over-smoothing problem (Chen et al., 2020a; Yan et al., 2022) when stacking deeply. Therefore, most existing GNNs are shallow networks, making the effects of model capacity mismatch negligible during GNN-to-MLP KD.

To explore the criteria for better GNN knowledge samples (nodes), we conduct an exploratory experiment to evaluate the roles played by GNN knowledge samples of different hardnesses during knowledge distillation. For example, we report in Fig. 2 the distillation accuracy of several representative methods for simple samples (bottom 50% hardness) and hard samples (top 50% hardness), as well as their overall accuracy. As can be seen from Fig. 2, those simple samples can be handled well by all methods, and the main difference in the performance of different distillation methods lies in their capability to handle those hard samples. In other words, *hard sample distillation may be a major performance bottleneck* of existing distillation algorithms. For example, FF-G2M improves the overall accuracy by 1.86% compared to GLNN, where hard samples contribute 3.27%, but simple samples contribute only 0.45%. Note that this phenomenon also exists in human education, where simple knowledge can be easily grasped by all students and therefore teachers are encouraged to spend more efforts in teaching hard knowledge. Therefore, we believe that not only should we not ignore those hard samples, but we should provide them with more supervision in a hardness-based manner.

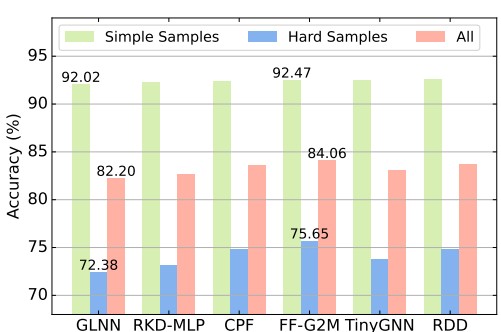

Figure 2: Classification accuracy of several distillation baselines for simple and hard samples.

### 3.3 HARDNESS-AWARE GNN-TO-MLP KNOWLEDGE DISTILLATION

One previous work (Zhou et al., 2021) defined knowledge hardness as the cross entropy on labeled data and proposed to weigh the distillation losses among samples in a hardness-based manner. To extend it to the transductive setting for graphs in this paper, we adopt the information entropy in Eq. (2) instead of the cross entropy as the knowledge hardness, and propose a variant of it as follows

$$\mathcal{L}_{\text{KD}} = \frac{1}{|\mathcal{V}|} \sum_{i \in \mathcal{V}} \left( 1 - e^{-\mathcal{H}(\mathbf{h}_i)/\mathcal{H}(\mathbf{z}_i)} \right) \cdot \mathcal{D}_{KL}\Big( \sigma\left(\mathbf{z}_i/\tau\right), \sigma\left(\mathbf{h}_i/\tau\right) \Big). \tag{3}$$

As far as GNN knowledge hardness is concerned, Eq. (3) reduces the weights of those hard samples with large higher $\mathcal{H}(\mathbf{z}_i)$, while leaving those simple samples to dominate the optimization. However, Sec. 3.2 shows that not only should we not ignore those hard samples, but we should pay more attention to them by providing more supervision. To this end, we propose a novel GNN-to-MLP KD framework, namely HGMD, which extracts a hardness-aware subgraph (the harder, the larger) for each sample separately and then distills the *subgraph-level knowledge* into the corresponding nodes of student MLPs. A high-level overview of the proposed HGMD framework is shown in Fig. 3.

#### 3.3.1 HARDNESS-AWARE SUBGRAPH EXTRACTION

We estimate the *distillation hardness* based on the *knowledge hardness* of both the teacher and the student, and then model the probability that the neighbors of a target node are included in the corresponding subgraph based on the distillation hardness. Intuitively, for any given node $v_i$, four key factors that influence the distillation hardness and subgraph size should be considered, including

- A harder sample with higher $\mathcal{H}(\mathbf{z}_i)$ in teacher GNNs should be assigned a larger subgraph.
- A sample with high uncertainty $\mathcal{H}(\mathbf{h}_i)$ in student MLPs requires a larger subgraph.
- A node $v_j \in \mathcal{N}_i$ with lower $\mathcal{H}(\mathbf{z}_j)$ has a higher probability to be included in the subgraph.
- Nodes in the subgraph are expected to share similar label distributions with the target node $v_i$.

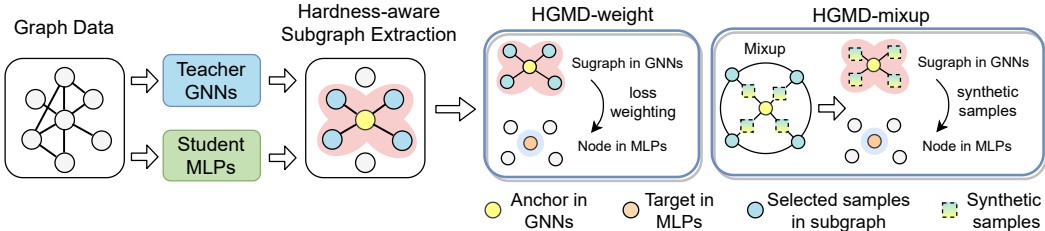

Figure 3: Illustration of the hardness-aware GNN-to-MLP distillation (HGMD) framework, which consists of hardness-aware subgraph extraction and two subgraph-level distillation schemes.

To satisfy these four properties, we model the probability $p_{j \to i}$ that a neighboring node $v_j \in \mathcal{N}_i$ of the target node $v_i$ is included in the subgraph based on the distillation hardness $r_{j \to i}$, as follow

$$p_{j \to i} = 1 - r_{j \to i}, \text{ where } r_{j \to i} = \exp\left(-\eta \cdot \mathcal{D}(\mathbf{z}_i, \mathbf{z}_j) \cdot \frac{\sqrt{\mathcal{H}(\mathbf{h}_i) \cdot \mathcal{H}(\mathbf{z}_i)}}{\mathcal{H}(\mathbf{z}_j)}\right) \in (0, 1] \quad (4)$$

where $\mathcal{D}(\mathbf{z}_i, \mathbf{z}_j)$ denotes the cosine similarity between $\mathbf{z}_i$ and $\mathbf{z}_j$, and we specify that $p_{i \to i} = 1$. In addition, $\eta$ is a hyperparameter that is used to control the overall hardness sensitivity. In this paper, we adopt an exponentially decaying strategy to set the hyperparameter $\eta$. Extensive qualitative and quantitative experiments are provided in Sec. 4.4 to demonstrate the effectiveness of our design.

### 3.3.2 HGMD-WEIGHT

Based on the sampling probabilities modeled in Eq. (4), we can easily sample a hardness-aware subgraph $g_i$ with node set $\mathcal{V}_i^g = \{v_j \sim \text{Bernoulli}(p_{j \to i}) \mid j \in (\mathcal{N}_i \cup i)\}$ for each target node $v_i$ by Bernoulli sampling. Next, a key issue being left is how to distill the subgraph-level knowledge from teacher GNNs into the corresponding nodes of student MLPs. A straightforward idea is to follow Wu et al. (2023a) to perform many-to-one knowledge distillation by optimizing the objective,

$$\mathcal{L}_{\text{KD}}^{\text{weight}} = \frac{1}{|\mathcal{V}|} \sum_{i \in \mathcal{V}} \frac{1}{|\mathcal{V}_i^g|} \sum_{i \in \mathcal{V}_i^g} p_{j \to i} \cdot \mathcal{D}_{KL}\left(\sigma\left(\mathbf{z}_j / \tau\right), \sigma\left(\mathbf{h}_i / \tau\right)\right). \quad (5)$$

Compared to the loss weighting of Eq. (3), the strengths of the HGMD-weight in Eq. (5) are four-fold: (1) it extends knowledge distillation from node-to-node single-teacher KD to subgraph-to-node multi-teacher KD, which introduces additional supervision; (2) it provides more supervision (i.e., larger subgraphs) for hard samples in a hardness-aware manner, rather than neglecting them by reducing their loss weights; (3) it inherits the benefit of loss weighting by assigning a large weight $p_{j \to i}$ to a sample $v_j$ with low hardness $\mathcal{H}(\mathbf{z}_j)$ in the subgraph; (4) it takes into account not only the knowledge hardness of the target node but also the nodes in the subgraph and their similarities to the target, enjoying more contextual information. While the modification from Eq. (3) to Eq. (5) does not introduce any additional parameters, it achieves a huge improvement, as shown in Table. 1.

### 3.3.3 HGMD-MIXUP

Recently, mixup (Abu-El-Haija et al., 2019), as an important data augmentation technique, has achieved great success. Combining mixup with our HGMD framework enables the generation of more GNN knowledge variants as additional supervision for those hard samples, which may help to improve the generalizability of the distilled student model. Inspired by this, we propose a **hardness-aware mixup** scheme to distill the subgraph-level knowledge from GNNs into MLPs. Instead of mixing the samples randomly, we mix them by emphasizing the sample with a high probability $p_{j \to i}$. Formally, for each target sample $v_i$, a synthetic sample $\boldsymbol{u}_{i,j}$ ($v_j \in \mathcal{V}_i^g$) will be generated by

$$\boldsymbol{u}_{i,j} = \lambda \cdot p_{j \to i} \cdot \mathbf{z}_j + (1 - \lambda \cdot p_{j \to i}) \cdot \mathbf{z}_i, \quad \lambda \sim \text{Beta}(\alpha, \alpha), \quad (6)$$

where $\text{Beta}(\alpha, \alpha)$ is a beta distribution parameterized by $\alpha$. For a node $v_j \in \mathcal{V}_i^g$ in the subgraph with lower hardness $\mathcal{H}(\mathbf{z}_j)$ and higher similarity $\mathcal{D}(\mathcal{H}(\mathbf{z}_i), \mathcal{H}(\mathbf{z}_j))$, the synthetic sample $\boldsymbol{u}_{i,j}$ will be closer to $\mathbf{z}_j$. Finally, we can distill the knowledge of synthetic samples $\{\boldsymbol{u}_{i,j}\}_{v_j \in \mathcal{V}_i^g}$ in the subgraph $g_i$ into the corresponding node $v_i$ of student MLPs by optimizing the objective, as follows

$$\mathcal{L}_{\text{KD}}^{\text{mixup}} = \frac{1}{|\mathcal{V}|} \sum_{i \in \mathcal{V}} \frac{1}{|\mathcal{V}_i^g|} \sum_{i \in \mathcal{V}_i^g} \mathcal{D}_{KL}\left(\sigma\left(\boldsymbol{u}_{i,j} / \tau\right), \sigma\left(\mathbf{h}_i / \tau\right)\right). \quad (7)$$

Compared to the weighting-based scheme (HGMD-weight) of Eq. (5), the mixup-based scheme (HGMD-mixup) generates more variants of GNN knowledge through data augmentation, which is more in line with our original intention of providing more supervision for hard sample distillation.

### 3.4 TRAINING STRATEGY

To achieve GNN-to-MLP knowledge distillation, we first pre-train the teacher GNNs with the classification loss $\mathcal{L}_{\text{label}} = \frac{1}{|\mathcal{V}_L|} \sum_{i \in \mathcal{V}_L} \text{CE}\left(y_i, \sigma(\mathbf{z}_i)\right)$, where $\text{CE}(\cdot)$ denotes the cross-entropy loss. Finally, we distill knowledge from teacher GNNs into student MLPs by the following objective,

$$\mathcal{L}_{\text{total}} = \frac{\beta}{|\mathcal{V}_L|} \sum_{i \in \mathcal{V}_L} \text{CE}\left(y_i, \sigma(\mathbf{h}_i)\right) + \left(1 - \beta\right) \mathcal{L}_{\text{KD}}, \tag{8}$$

where $\beta$ is the hyperparameter to trade-off the classification and distillation losses. Besides, the pseudo-code of HGMD (taking HGMD-mixup as an example) is summarized in Algorithm. 1.

---

**Algorithm 1** Algorithm for the *Hardness-aware GNN-to-MLP Distillation* (MGMD-mixup)

**Input:** Feature Matrix: $\mathbf{X}$; Adjacency Matrix: $\mathbf{A}$; Number of Epochs: $E$.
**Output:** Predicted Labels $\mathcal{Y}_U$ and network parameters of the distilled student MLPs $f_\gamma^{\mathcal{S}}(\cdot)$.
1: Randomly initialize the parameters of teacher GNNs $f_\theta^{\mathcal{T}}(\cdot)$ and student MLPs $f_\gamma^{\mathcal{S}}(\cdot)$.
2: Compute node embeddins $\{\mathbf{z}_i\}_{i=1}^N$ of GNNs and pre-train GNNs until convergence by $\mathcal{L}_{\text{label}}$.
3: **for** $epoch \in \{1, 2, \cdots, E\}$ **do**
4:     Compute node embeddins $\{\mathbf{h}_i\}_{i=1}^N$ of the student MLPs.
5:     Calculate probabilities $\{p_{j \to i}\}_{i \in \mathcal{V}, j \in \mathcal{N}_i}$ and extract hardness-aware subgraphs $\{g_i\}_{i=1}^N$.
6:     Generate synthetic samples $\{\boldsymbol{u}_{i,j}\}_{i \in \mathcal{V}, j \in \mathcal{V}_i^g}$ by hardness-aware mixup in Eq. (6).
7:     Calculate mixup-based knowledge distillation loss $\mathcal{L}_{\text{KD}}^{\text{mixup}}$ by Eq. (7).
8:     Update MLP parameters $f_\gamma^{\mathcal{S}}(\cdot)$ by back propagation of the total loss $\mathcal{L}_{\text{total}}$ by Eq. (8).
9: **end for**
10: **return** Predicted labels $\mathcal{Y}_U$ for unlabeled data and parameters $f_\gamma^{\mathcal{S}}(\cdot)$ of the student MLPs.

---

### 3.5 ANALYSIS OF MODEL PARAMETERS AND COMPUTATIONAL COMPLEXITY

Compared to vanilla GNN-to-MLP KD, such as GLNN (Zhang et al., 2022), HGMD does not introduce any additional learnable parameters in the process of subgraph extraction and subgraph distillation. In other words, our method is almost non-parametric. In terms of the computational complexity, the time complexity of HGMD mainly comes from two parts: (1) GNN training $\mathcal{O}(|\mathcal{V}|dF + |\mathcal{E}|F)$ and (2) Knowledge distillation $\mathcal{O}(|\mathcal{E}|F)$, where $d$ and $F$ are the dimensions of input and hidden spaces. The total time complexity $\mathcal{O}(|\mathcal{V}|dF + |\mathcal{E}|F)$ is linear w.r.t the number of nodes $|\mathcal{V}|$ and edges $|\mathcal{E}|$. This indicates that the time complexity of knowledge distillation in HGMD is basically on par with GNN training and does not suffer from an overly high computational burden. A comparison of HGMD with other methods in terms of running time can be found in **Appendix E**.

## 4 EXPERIMENTS

### 4.1 EXPERIMENTAL SETUP

In this paper, we evaluate HGMD on *eight* real-world datasets, including Cora (Sen et al., 2008), Citeseer (Giles et al., 1998), Pubmed (McCallum et al., 2000), Coauthor-CS, Coauthor-Physics, Amazon-Photo (Shchur et al., 2018), ogbn-arxiv (Hu et al., 2020), and ogbn-products (Hu et al., 2020). A statistical overview of these datasets is available in **Appendix A**. Besides, we defer the implementation details and hyperparameter settings for each dataset to **Appendix B**. In addition, we consider three common GNN architectures as GNN teachers, including GCN (Kipf & Welling, 2016), GraphSAGE (Hamilton et al., 2017), and GAT (Veličković et al., 2017), and comprehensively evaluate two distillation schemes, HGMD-weight and HGMD-mixup, respectively. Furthermore, we also compare HGMD with two types of state-of-the-art graph distillation methods, including (1) GNN-to-GNN KD: (Yang et al., 2020), TinyGNN (Yan et al., 2020), GraphAKD (He et al., 2022), RDD (Zhang et al., 2020b), FreeKD (Feng et al., 2022), and GNN-SD (Chen et al., 2020b); and (2) GNN-to-MLP KD: CPF (Yang et al., 2021), RKD-MLP (Anonymous, 2023), GLNN (Zhang et al., 2022), FF-G2M (Wu et al., 2023a), NOSMOG (Tian et al., 2023), and KRD (Wu et al., 2023b).

Table 1: Accuracy $\pm$ std (%) on seven datasets, where three different GNN architectures (GCN, GraphSAGE, and GAT) are considered as the teacher models. The best metrics are marked by **bold**.

| Teacher | Student | Cora | Citeseer | Pubmed | Photo | CS | Physics | ogbn-arxiv | Average |
|---|---|---|---|---|---|---|---|---|---|
| MLPs | - | $59.58_{\pm0.97}$ | $60.32_{\pm0.61}$ | $73.40_{\pm0.68}$ | $78.65_{\pm1.68}$ | $87.82_{\pm0.64}$ | $88.81_{\pm1.08}$ | $54.63_{\pm0.84}$ | - |
| GCN | - | $81.70_{\pm0.96}$ | $71.64_{\pm0.34}$ | $79.48_{\pm0.21}$ | $90.63_{\pm1.53}$ | $90.00_{\pm0.58}$ | $92.45_{\pm0.53}$ | $71.20_{\pm0.17}$ | - |
| | GLNN | $82.20_{\pm0.73}$ | $71.72_{\pm0.30}$ | $80.16_{\pm0.20}$ | $91.42_{\pm1.61}$ | $92.22_{\pm0.72}$ | $93.11_{\pm0.39}$ | $67.76_{\pm0.23}$ | - |
| | Loss-Weighting | $83.25_{\pm0.69}$ | $72.98_{\pm0.41}$ | $81.20_{\pm0.50}$ | $91.76_{\pm1.52}$ | $93.16_{\pm0.66}$ | $93.46_{\pm0.43}$ | $68.56_{\pm0.27}$ | - |
| | HGMD-weight | $84.42_{\pm0.54}$ | $74.42_{\pm0.50}$ | $81.86_{\pm0.44}$ | $92.94_{\pm1.37}$ | $93.93_{\pm0.33}$ | $94.09_{\pm0.56}$ | $70.76_{\pm0.19}$ | - |
| | *Improv.* | 2.22 | 2.70 | 1.70 | 1.52 | 1.71 | 0.98 | 3.00 | 1.98 |
| | HGMD-mixup | $\mathbf{84.66}_{\pm0.47}$ | $\mathbf{74.62}_{\pm0.40}$ | $\mathbf{82.02}_{\pm0.45}$ | $\mathbf{93.33}_{\pm1.31}$ | $\mathbf{94.16}_{\pm0.32}$ | $\mathbf{94.27}_{\pm0.63}$ | $\mathbf{71.09}_{\pm0.21}$ | - |
| | *Improv.* | 2.46 | 2.90 | 1.86 | 1.91 | 1.94 | 1.16 | 3.33 | 2.22 |
| GraphSAGE | - | $82.02_{\pm0.94}$ | $71.76_{\pm0.49}$ | $79.36_{\pm0.45}$ | $90.56_{\pm1.69}$ | $89.29_{\pm0.77}$ | $91.97_{\pm0.91}$ | $71.06_{\pm0.27}$ | - |
| | GLNN | $81.86_{\pm0.88}$ | $71.52_{\pm0.54}$ | $80.32_{\pm0.38}$ | $91.34_{\pm1.46}$ | $92.00_{\pm0.57}$ | $92.82_{\pm0.93}$ | $68.30_{\pm0.19}$ | - |
| | Loss-Weighting | $83.16_{\pm0.76}$ | $72.30_{\pm0.47}$ | $80.92_{\pm0.46}$ | $91.63_{\pm1.31}$ | $92.84_{\pm0.60}$ | $93.28_{\pm0.72}$ | $69.04_{\pm0.22}$ | - |
| | HGMD-weight | $84.36_{\pm0.60}$ | $\mathbf{73.70}_{\pm0.50}$ | $81.50_{\pm0.57}$ | $93.01_{\pm1.19}$ | $93.77_{\pm0.47}$ | $\mathbf{94.21}_{\pm0.57}$ | $71.62_{\pm0.26}$ | - |
| | *Improv.* | 2.50 | 2.18 | 1.18 | 1.67 | 1.77 | 1.39 | 3.32 | 2.00 |
| | HGMD-mixup | $\mathbf{84.54}_{\pm0.53}$ | $73.48_{\pm0.53}$ | $\mathbf{81.66}_{\pm0.36}$ | $\mathbf{93.29}_{\pm1.22}$ | $\mathbf{94.03}_{\pm0.43}$ | $94.12_{\pm0.61}$ | $\mathbf{71.86}_{\pm0.24}$ | - |
| | *Improv.* | 2.68 | 1.96 | 1.34 | 1.95 | 2.03 | 1.30 | 3.52 | 2.11 |
| GAT | - | $81.66_{\pm1.04}$ | $70.78_{\pm0.60}$ | $79.88_{\pm0.85}$ | $90.06_{\pm1.38}$ | $90.90_{\pm0.37}$ | $91.97_{\pm0.58}$ | $71.08_{\pm0.19}$ | - |
| | GLNN | $81.78_{\pm0.75}$ | $70.96_{\pm0.86}$ | $80.48_{\pm0.47}$ | $91.22_{\pm1.45}$ | $92.44_{\pm0.41}$ | $92.70_{\pm0.56}$ | $68.56_{\pm0.22}$ | - |
| | Loss-Weighting | $82.69_{\pm0.74}$ | $71.80_{\pm0.52}$ | $81.27_{\pm0.55}$ | $91.58_{\pm1.42}$ | $92.96_{\pm0.58}$ | $93.10_{\pm0.64}$ | $69.32_{\pm0.25}$ | - |
| | HGMD-weight | $\mathbf{84.22}_{\pm0.77}$ | $73.10_{\pm0.83}$ | $82.02_{\pm0.59}$ | $93.18_{\pm1.96}$ | $94.09_{\pm1.33}$ | $\mathbf{94.29}_{\pm0.56}$ | $71.76_{\pm0.26}$ | - |
| | *Improv.* | 2.44 | 2.14 | 1.54 | 1.96 | 1.65 | 1.59 | 3.20 | 2.07 |
| | HGMD-mixup | $84.02_{\pm0.65}$ | $\mathbf{73.18}_{\pm0.79}$ | $\mathbf{82.16}_{\pm0.64}$ | $\mathbf{93.43}_{\pm1.26}$ | $\mathbf{94.20}_{\pm0.27}$ | $94.19_{\pm0.43}$ | $\mathbf{72.31}_{\pm0.20}$ | - |
| | *Improv.* | 2.24 | 2.22 | 1.68 | 2.21 | 1.76 | 1.49 | 3.75 | 2.19 |

## 4.2 COMPARATIVE RESULTS

To evaluate the effectiveness of the HGMD framework, we compare its two instantiations, HGMD-weight and HGMD-mixup, with GLNN of Eq. (1) and Loss-Weighting of Eq. (3), respectively. The experiments are conducted on seven datasets with three different GNN architectures as teacher GNNs, where *imporv.* denotes the performance improvements with respect to GLNN. From the results reported in Table. 1, we can make three observations: (1) Both HGMD-weight and HGMD-mixup perform much better than vanilla MLP, GLNN, and Loss-Weighting on all seven datasets, especially on the large-scale ogbn-arxiv dataset. (2) Both HGMD-weight and HGMD-mixup are applicable to various types of teacher GNN architectures. For example, HGMD-mixup outperforms GLNN by 2.22% (GCN), 2.11% (SAGE), and 2.19% (GAT) averaged over seven datasets, respectively. (3) Overall, HGMD-mixup performs slightly better than HGMD-weight across various datasets and GNN architectures, owing to more knowledge variants augmented by the mixup.

Furthermore, we compare HGMD-weight and HGDM-mixup with several state-of-the-art graph distillation methods, including both GNN-to-GNN and GNN-to-MLP KD. The experimental results reported in Table. 2 show that (1) Despite being completely non-parametric methods, HGMD-weight and HGMD-mixup both perform much better than existing GNN-to-MLP baselines on 5 out of 8 datasets. (2) HGMD-weight and HGMD-mixup outperform those GNN-to-GNN baselines on four relatively small datasets (i.e., Cora, Citeseer, Pubmed, and Photo). Besides, their performance is comparable to those GNN-to-GNN baselines on four relatively large datasets (i.e., CS, Physics, and ogbn-arxiv, products). These observations indicate that distilled MLPs have the same expressive potential as teacher GNNs, and that "parametric" is not a must for knowledge distillation. In addition, we have also evaluated HGMD in the **inductive setting** and the results are provided in **Appendix C**.

## 4.3 ABLATION STUDY

To evaluate how hardness-aware subgraph extraction (SubGraph) and two subgraph distillation strategies (weight and mixup) influence performance, we compare vanilla GCNs and GLNN with the following five schemes: (A) *Subgraph-only*: extract hardness-aware subgraphs and then distill their knowledge into the student MLP with equal loss weights; (B) *Weight-only*: take the full neighborhoods as subgraphs and then distill by hardness-aware weighting as in Eq. (5); (C) *Mixup-only*: take the full neighborhoods as subgraphs and then distill by hardness-aware mixup as in Eq. (7); (D) *HGMD-weight*; and (E) *HGMD-mixup*. We can observe from the experimental results reported in Table. 3 that (1) SubGraph plays a very important role in improving performance, which illustrates the benefits of performing knowledge distillation at the subgraph level compared to the node level, as it provides more supervision for those hard samples in a hardness-aware manner. (2) Both hardness-aware weighting and mixup help improve performance, especially the latter. (3) Combin-

Table 2: Accuracy $\pm$ std (%) of various graph knowledge distillation algorithms in the *transductive* setting on eight datasets, where **bold** and underline denote the best and second metrics, respectively.

| Category | Method | Cora | Citeseer | Pubmed | Photo | CS | Physics | ogbn-arxiv | products |
|---|---|---|---|---|---|---|---|---|---|
| Vanilla | MLPs | $59.58_{\pm0.97}$ | $60.32_{\pm0.61}$ | $73.40_{\pm0.68}$ | $78.65_{\pm1.68}$ | $87.82_{\pm0.64}$ | $88.81_{\pm1.08}$ | $54.63_{\pm0.84}$ | $61.89_{\pm0.18}$ |
| | Vanilla GCNs | $81.70_{\pm0.96}$ | $71.64_{\pm0.34}$ | $79.48_{\pm0.21}$ | $90.63_{\pm1.53}$ | $90.00_{\pm0.58}$ | $92.45_{\pm0.53}$ | $71.20_{\pm0.17}$ | $75.42_{\pm0.28}$ |
| GNN-to-GNN | LSP | $82.70_{\pm0.43}$ | $72.68_{\pm0.62}$ | $80.86_{\pm0.50}$ | $91.74_{\pm1.42}$ | $92.56_{\pm0.45}$ | $92.85_{\pm0.46}$ | $71.57_{\pm0.25}$ | $74.18_{\pm0.41}$ |
| | GNN-SD | $82.54_{\pm0.36}$ | $72.34_{\pm0.55}$ | $80.52_{\pm0.37}$ | $91.83_{\pm1.58}$ | $91.92_{\pm0.51}$ | $93.22_{\pm0.66}$ | $70.90_{\pm0.23}$ | $73.90_{\pm0.23}$ |
| | GraphAKD | $83.71_{\pm0.77}$ | $72.68_{\pm0.71}$ | $80.96_{\pm0.39}$ | - | - | - | - | - |
| | TinyGNN | $83.10_{\pm0.53}$ | $73.24_{\pm0.72}$ | $81.20_{\pm0.44}$ | $92.03_{\pm1.49}$ | $93.78_{\pm0.38}$ | $93.70_{\pm0.56}$ | $72.18_{\pm0.27}$ | $74.76_{\pm0.30}$ |
| | RDD | $83.68_{\pm0.40}$ | $73.64_{\pm0.50}$ | $81.74_{\pm0.44}$ | $92.18_{\pm1.45}$ | $\mathbf{94.20}_{\pm0.48}$ | $\underline{94.14}_{\pm0.39}$ | $\underline{72.34}_{\pm0.17}$ | $75.30_{\pm0.24}$ |
| | FreeKD | $83.84_{\pm0.47}$ | $73.92_{\pm0.47}$ | $81.48_{\pm0.38}$ | $92.38_{\pm1.54}$ | $93.65_{\pm0.43}$ | $93.87_{\pm0.48}$ | $\mathbf{72.50}_{\pm0.29}$ | $75.84_{\pm0.25}$ |
| GNN-to-MLP | GLNN | $82.20_{\pm0.73}$ | $71.72_{\pm0.30}$ | $80.16_{\pm0.20}$ | $91.42_{\pm1.61}$ | $92.22_{\pm0.72}$ | $93.11_{\pm0.39}$ | $67.76_{\pm0.23}$ | $65.18_{\pm0.27}$ |
| | CPF | $83.56_{\pm0.48}$ | $72.98_{\pm0.60}$ | $81.54_{\pm0.47}$ | $91.70_{\pm1.50}$ | $93.42_{\pm0.48}$ | $93.47_{\pm0.41}$ | $69.05_{\pm0.18}$ | $68.80_{\pm0.24}$ |
| | RKD-MLP | $82.68_{\pm0.45}$ | $73.42_{\pm0.45}$ | $81.32_{\pm0.32}$ | $91.28_{\pm1.48}$ | $93.16_{\pm0.64}$ | $93.26_{\pm0.37}$ | $69.87_{\pm0.25}$ | $72.52_{\pm0.35}$ |
| | FF-G2M | $84.06_{\pm0.43}$ | $73.85_{\pm0.51}$ | $81.62_{\pm0.37}$ | $91.84_{\pm1.42}$ | $93.35_{\pm0.55}$ | $93.59_{\pm0.43}$ | $69.64_{\pm0.26}$ | $71.69_{\pm0.31}$ |
| | NOSMOG | $83.80_{\pm0.50}$ | $74.08_{\pm0.45}$ | $81.49_{\pm0.53}$ | $\underline{93.18}_{\pm1.20}$ | $93.54_{\pm0.98}$ | $93.61_{\pm0.58}$ | $71.20_{\pm0.24}$ | $\underline{76.14}_{\pm0.32}$ |
| | HGMD-weight | $\underline{84.42}_{\pm0.54}$ | $\underline{74.42}_{\pm0.50}$ | $\underline{81.86}_{\pm0.44}$ | $92.94_{\pm1.37}$ | $93.93_{\pm0.33}$ | $94.09_{\pm0.56}$ | $70.76_{\pm0.19}$ | $75.21_{\pm0.22}$ |
| | HGMD-mixup | $\mathbf{84.66}_{\pm0.47}$ | $\mathbf{74.62}_{\pm0.40}$ | $\mathbf{82.02}_{\pm0.45}$ | $\mathbf{93.33}_{\pm1.31}$ | $\underline{94.16}_{\pm0.32}$ | $\mathbf{94.27}_{\pm0.63}$ | $71.09_{\pm0.21}$ | $\mathbf{76.25}_{\pm0.18}$ |

Table 3: Ablation study on the hardness-aware subgraph extraction and distillation modules.

| Scheme | KD | SubGraph | Weight | Mixup | Cora | Citeseer | Pubmed | Photo | CS | Physics |
|---|---|---|---|---|---|---|---|---|---|---|
| Vinilla GCNs | ✗ | ✗ | ✗ | ✗ | $81.70_{\pm0.96}$ | $71.64_{\pm0.34}$ | $79.48_{\pm0.21}$ | $90.63_{\pm1.53}$ | $90.00_{\pm0.58}$ | $92.45_{\pm0.53}$ |
| GLNN | ✓ | | | | $82.20_{\pm0.73}$ | $71.72_{\pm0.30}$ | $80.16_{\pm0.20}$ | $91.42_{\pm1.61}$ | $92.22_{\pm0.72}$ | $93.11_{\pm0.39}$ |
| SubGraph-only | ✓ | ✓ | | | $83.78_{\pm0.55}$ | $74.26_{\pm0.69}$ | $81.46_{\pm0.45}$ | $91.46_{\pm1.37}$ | $93.48_{\pm0.53}$ | $93.80_{\pm0.63}$ |
| Weight-only | ✓ | | ✓ | | $83.56_{\pm0.72}$ | $73.96_{\pm0.46}$ | $81.18_{\pm0.64}$ | $92.14_{\pm1.37}$ | $93.23_{\pm0.46}$ | $93.50_{\pm0.67}$ |
| Mixup-only | ✓ | | | ✓ | $83.90_{\pm0.73}$ | $74.14_{\pm0.50}$ | $81.34_{\pm0.38}$ | $92.40_{\pm1.26}$ | $93.66_{\pm0.52}$ | $93.68_{\pm0.70}$ |
| HGMD-weight | ✓ | ✓ | ✓ | | $84.42_{\pm0.54}$ | $74.42_{\pm0.50}$ | $81.86_{\pm0.44}$ | $92.94_{\pm1.37}$ | $93.93_{\pm0.33}$ | $94.09_{\pm0.56}$ |
| HGMD-mixup | ✓ | ✓ | | ✓ | $\mathbf{84.66}_{\pm0.47}$ | $\mathbf{74.62}_{\pm0.40}$ | $\mathbf{82.02}_{\pm0.45}$ | $\mathbf{93.33}_{\pm1.31}$ | $\mathbf{94.16}_{\pm0.32}$ | $\mathbf{94.27}_{\pm0.63}$ |

ing the two different designs (subgraph extraction and subgraph distillation) together can further improve performance on top of each on all six datasets. Furthermore, we provide a comparison between HGMD-mixup and other baselines in terms of robustness to feature noise in **Appendix D**.

### 4.4 QUALITATIVE AND QUANTITATIVE ANALYSIS ON HARDNESS AWARENESS

**Case Study of Hardness-aware Subgraphs.** To intuitively show what "hardness awareness" means, we select three GNN knowledge samples with different hardness levels from four datasets, respectively. Next, we mark the hardness of each knowledge sample as well as their neighboring nodes according to the color bar on their right side, where a darker blue indicates a higher distillation hardness. In addition, we use the edge color to denote the probability of the corresponding neighboring node being sampled into the hardness-aware subgraph, according to another color bar displayed on the right. We can observe from Fig. 4 that: (1) For a given target node, neighboring nodes with lower hardness (lighter blue) tend to have a higher probability of being sampled into the subgraph. (2) A target node with higher hardness (darker blue) has a higher probability of having its neighboring nodes sampled. In other words, *the sampling probability of neighboring nodes is actually a trade-off between their own hardness and the hardness of the target node*. For example, when the hardness of the target node is 0.36, a neighboring node with a hardness of 0.61 is still hard to be sampled, as shown in Fig. 4(a); however, when the hardness of the target node is 1.73, even a neighboring node with a hardness of 1.52 has a high sampling probability, which is close to 0.7.

**3D Histogram on Hardness and Similarity.** We show in Fig. 5(a) and Fig. 5(b) the 3D histograms of the sampling probability of neighboring nodes w.r.t their hardness, their cosine similarity to the target node, and the hardness of the target node, from which we can observe that: (1) As the hardness of a target node increases, the sampling probability of its neighboring nodes also increases; (2) Neighboring nodes with lower hardness have a higher probability of being sampled into the subgraph; (3) As the cosine similarity between neighboring nodes and target node increases, their sampling probability also increases; However, when the hardness of the target node is high, an overly high similarity means that the hardness of neighboring nodes will also be high, which in turn reduces the sampling probability, which is actually a trade-off between high similarity and low hardness.

**Training Curves.** We report in Fig. 5(c) the average entropy of the nodes in student MLPs and the average size of sampled subgraphs during training on the Cora dataset. It can be seen that *there*

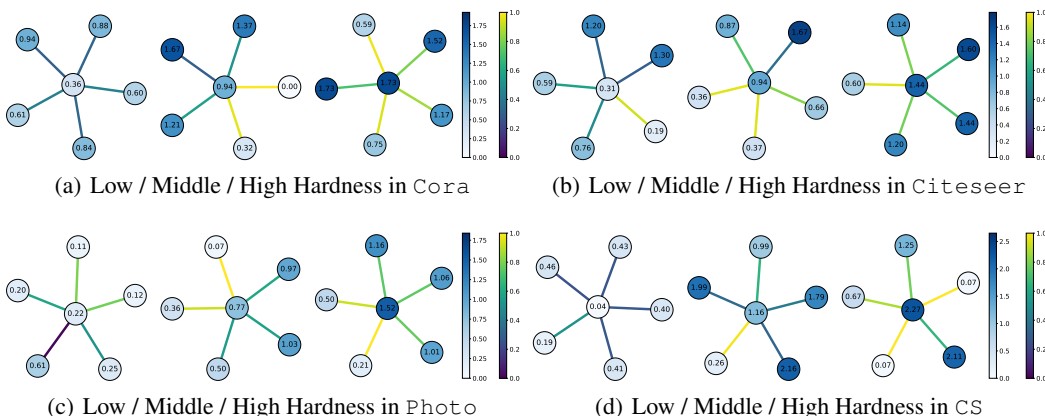

Figure 4: Visualizations of three GNN knowledge samples of different hardness levels (Low / Middle / High) on four datasets, where the node and edge colors indicate the hardness of knowledge samples and the sampling probability of neighboring nodes, respectively, and the color bars are on the right.

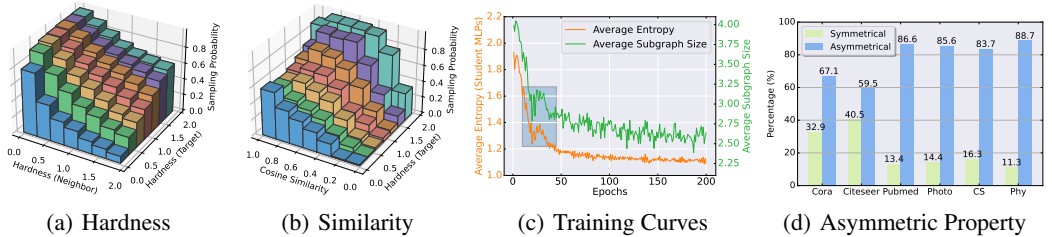

Figure 5: *(ab)* 3D histogram of the sampling probability of neighboring nodes w.r.t their hardness, hardness of the target node, and their cosine similarity to the target on Cora. *(c)* Training curves for the average entropy of student MLPs and the average size of sampled subgraphs on Cora. *(d)* Ratio of connected nodes sampled symmetrically and asymmetrically among all edges on six datasets.

*exists a resonance between the two curves*. As the training progresses, the uncertainty of the student MLPs decreases and thus additional supervision required for distillation can be reduced accordingly.

**Asymmetric Property of Subgraph Extraction.** We statistically calculate the ratios of two connected nodes among all edges that ***are*** and ***are not*** sampled into each other's subgraphs simultaneously, called symmetrized and asymmetrized sampling. The histogram in Fig. 5(d) shows that subgraph extraction is mostly asymmetric, especially for large-scale datasets. This is because our subgraph extraction is performed in a hardness-aware manner, where low-hardness neighboring nodes of a high-hardness target node have a higher sampling probability, but not vice versa. We believe that such asymmetric property of subgraph extraction is an important aspect of the effectiveness of our HGMD, since it essentially transforms an undirected graph into a directed graph for processing.

## 5 CONCLUSION

In this paper, we explore thoroughly why *"better teacher, better student"* does not hold true for GNN-to-MLP KD from the perspective of **hardness** rather than **correctness**. We identify that hard sample distillation may be a major performance bottleneck of existing distillation algorithms. To address this problem, we propose a novel *Hardness-aware GNN-to-MLP Distillation* (HGMD) framework, which distills knowledge from teacher GNNs at the subgraph level (rather than the node level) in a hardness-aware manner to provide more supervision for those hard samples. Extensive experiments demonstrate the superiority of HGMD across various datasets and GNN architectures. Limitations still exist, for example, designing better hardness metrics or introducing additional learnable parameters for knowledge distillation may be promising research directions for future work.

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

APPENDIX

A. DATASET STATISTICS

*Eight* real-world graph datasets are used to evaluate the proposed HGMD framework. An overview summary of the dataset characteristics is given in Table. A1. For the three small-scale datasets, including Cora, Citeseer, and Pubmed, we follow the data splitting strategy in (Kipf & Welling, 2016). For the three large-scale datasets, including Coauthor-CS, Coauthor-Physics, and Amazon-Photo, we follow Zhang et al. (2022); Yang et al. (2021) to randomly split the data into train/val/test sets, and each random seed corresponds to a different data splitting. For the ogbn-arxiv and ogbn-products datasets, we use the public data splits provided by the authors (Hu et al., 2020).

Table A1: Statistical information of the eight datasets.

| Dataset | Cora | Citeseer | Pubmed | Photo | CS | Physics | ogbn-arxiv | ogbn-products |
|---|---|---|---|---|---|---|---|---|
| # Nodes | 2,708 | 3,327 | 19,717 | 7,650 | 18,333 | 34,493 | 169,343 | 2,449,029 |
| # Edges | 5,278 | 4,614 | 44,324 | 119,081 | 81,894 | 247,962 | 1,166,243 | 61,859,140 |
| # Features | 1,433 | 3,703 | 500 | 745 | 6,805 | 8,415 | 128 | 100 |
| # Classes | 7 | 6 | 3 | 8 | 15 | 5 | 40 | 47 |

B. IMPLEMENTATION DETAILS

The following hyperparameters are set the same for all datasets: Epoch $E = 500$, learning rate $lr = 0.01$ (0.005 for ogbn-axriv), weight decay $decay =$ 5e-4 (0.0 for ogbn-arxiv), and layer number $L = 2$ (3 for Cora and ogbn-arxiv). The other dataset-specific hyperparameters are determined by an AutoML toolkit NNI with the search spaces as: hidden dimension $F = \{256, 512, 2048\}$, distillation temperature $\tau = \{0.8, 0.9, 1.0, 1.1, 1.2\}$, loss weight $\beta = \{0.0, 0.1, 0.2, 0.3, 0.4, 0.5\}$, coefficient of beta distribution $\alpha = \{0.3, 0.4, 0.5\}$. Moreover, the hyperparameter $\eta$ in Eq. (4) is initially set to $\{1, 5, 10\}$ and then decays exponentially with the decay step of 250. The experiments are implemented based on the DGL library (Wang et al., 2019) using the PyTorch 1.6.0 with Intel(R) Xeon(R) Gold 6240R @ 2.40GHz CPU and NVIDIA V100 GPU. For a fair comparison, the model with the highest validation accuracy will be selected for testing. Besides, each set of experiments is run five times with different random seeds, and the averages are reported as metrics.

For all baselines, we did not directly copy the results from their original papers but reproduced them by distilling from the same teacher GNNs as in this paper, under the same settings and data splits. As we know, the performance of the distilled student MLPs depends heavily on the quality of teacher GNNs. However, we have no way to get the checkpoints of the teacher models used in previous baselines, i.e., we cannot guarantee that the student MLPs in all baselines are distilled from the same teacher GNNs. For the purpose of a fair comparison, we have to train teacher GNNs from scratch and then reproduce the results of previous baselines by distilling the knowledge from the SAME teacher GNNs. Therefore, even if we follow the default implementation and hyperparameters of these baselines exactly, there is no way to get identical results on different hardware devices.

C. INDUCTIVE SETTING

We compare HGMD-mixup with vanilla GCNs, GLNN (Zhang et al., 2022), KRD (Wu et al., 2023b), and NOSMOG (Tian et al., 2023) in the inductive setting with GCNs as teacher GNNs. Considering the importance of node positional features (POS) in the inductive setting (as revealed by the ablation study in NOGMOG), we consider the performance of HGMD-mixup and NOGMOG w/ and w/o POS, respectively. We can find from the results in Table. A2 that (1) POS features play a crucial role, especially on the large-scale ogbn-arxiv dataset. (2) HGMD-mixup outperforms GLNN and KRD by a large margin, and is comparable to NOSMOG regardless of w/ and w/o POS features.

D. ROBUSTNESS EVALUATION

We follow Zhang et al. (2022); Tian et al. (2023) to evaluate the robustness of the model to feature noise by adding different levels of Gaussian noise to node features by replacing $\mathbf{X}$ with $\widetilde{\mathbf{X}} = (1 -$

Table A2: Acuracy $\pm$ std (%) in the *inductive* setting on seven datasets, where HGMD-mixup outperforms GLNN by a wide margin on all seven datasets and is comparable to NOSMOG and KRD.

| Teacher | Student | Cora | Citeseer | Pubmed | Photo | CS | Physics | ogbn-arxiv |
|---|---|---|---|---|---|---|---|---|
| | | **Inductive Setting** | | | | | | |
| MLPs | - | $59.20_{\pm1.26}$ | $60.16_{\pm0.87}$ | $73.26_{\pm0.83}$ | $79.02_{\pm1.42}$ | $87.90_{\pm0.58}$ | $89.10_{\pm0.90}$ | $54.46_{\pm0.52}$ |
| GCNs | - | $79.30_{\pm0.49}$ | $71.46_{\pm0.36}$ | $78.10_{\pm0.51}$ | $89.32_{\pm1.63}$ | $90.07_{\pm0.60}$ | $92.05_{\pm0.78}$ | $70.88_{\pm0.35}$ |
| | GLNN | $71.24_{\pm0.55}$ | $70.76_{\pm0.30}$ | $80.16_{\pm0.73}$ | $89.92_{\pm1.34}$ | $92.08_{\pm0.98}$ | $92.89_{\pm0.88}$ | $60.92_{\pm0.31}$ |
| | KRD | $73.78_{\pm0.55}$ | $71.80_{\pm0.41}$ | $81.48_{\pm0.29}$ | $90.37_{\pm1.79}$ | $93.15_{\pm0.43}$ | $93.86_{\pm0.55}$ | $62.85_{\pm0.32}$ |
| | NOSMOG (w/o POS) | $73.18_{\pm0.45}$ | $72.40_{\pm0.51}$ | $80.84_{\pm0.46}$ | $90.37_{\pm1.14}$ | $92.87_{\pm0.53}$ | $93.56_{\pm0.61}$ | $62.88_{\pm0.30}$ |
| | HGMD-mixup (w/o POS) | $\underline{73.92}_{\pm0.47}$ | $73.05_{\pm0.34}$ | $\mathbf{81.78}_{\pm0.59}$ | $91.10_{\pm1.59}$ | $\mathbf{93.65}_{\pm0.64}$ | $\underline{94.10}_{\pm0.70}$ | $63.20_{\pm0.28}$ |
| | NOSMOG (w/ POS) | $73.64_{\pm0.53}$ | $\underline{73.10}_{\pm0.47}$ | $81.32_{\pm0.38}$ | $\underline{91.26}_{\pm1.49}$ | $93.47_{\pm0.71}$ | $93.94_{\pm0.65}$ | $\mathbf{71.48}_{\pm0.35}$ |
| | HGMD-mixup (w/ POS) | $\mathbf{74.24}_{\pm0.31}$ | $\mathbf{73.25}_{\pm0.40}$ | $\underline{81.67}_{\pm0.46}$ | $\mathbf{91.32}_{\pm1.71}$ | $\underline{93.51}_{\pm0.54}$ | $\mathbf{94.44}_{\pm0.75}$ | $\underline{70.24}_{\pm0.48}$ |

$\alpha)\mathbf{X} + \alpha\mathbf{N}$, where $\mathbf{N}$ represents Gaussian noise that is independent from $\mathbf{X}$, and $\alpha \in [0, 1]$ incates the noise level. The results of GLNN, NOSMOG, and HGMD-mixup at 11 different noise ratios averaged over seven datasets (e.g., Cora, Citeseer, Pubmed, Photo, CS, Physics, and ogbn-arxiv) in the transductive setting are shown in Fig. A1. The results demonstrate the superior robustness of HGMD-mixup, especially at high noise ratios. This is thanks to the fact that our HGMD is hardness-aware, which can better identify those hard samples (usually with more feature noise), allowing the model to benefit more from those potential high-quality knowledge samples with low feature noise.

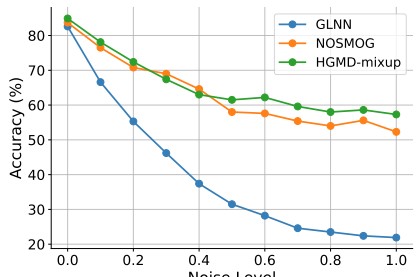

Figure A1: Robustness to Feature Noise.

| HGMD-mixup | Training Time (ms / epoch) | | |
|---|---|---|---|
| | GLNN | KRD | HGMD-mixup |
| Cora | 3.35 | 3.94 | 3.71 |
| Citeseer | 3.70 | 4.71 | 4.33 |
| Pubmed | 3.90 | 6.45 | 5.74 |
| Photo | 4.72 | 7.65 | 6.86 |
| CS | 18.03 | 23.74 | 21.75 |
| Physics | 34.16 | 43.15 | 39.51 |

Table A3: Training Time Analysis.

## E. RUNNING TIME ANALYSIS

All distillation algorithms have two parts: (1) training the teacher, and (2) teacher-to-student distillation. As a result, the computational complexity increase in training time is unavoidable. Indeed, distillation itself is more concerned with inference rather than training efficiency, aiming to shift considerable work from the latency-sensitive inference stage, where time reduction in milliseconds makes a huge difference, to the less latency-insensitive training stage, where time cost in hours or days is often tolerable. As a non-parametric approach, we believe that HGMD has an advantage in terms of training time compared to previous baselines. A comparison between GLNN (Zhang et al., 2022), KRD (Wu et al., 2023b), and HGMD-mixup in terms of running time (ms) per training epoch for knowledge distillation is shown in Table. A3, where we adopt 2-layer GCNs and MLPs with a hidden dimension of 256 as teacher and student models for GLNN, KRD, and HGMD-mixup. Due to the proposed subgraph-level knowledge distillation scheme, the running time of HGMD-mixup and KRD increases a bit on top of GLNN, but our HGMD-mixup still trains faster than KRD.

## F. EVALUATION OF KNOWLEDGE HARDNESS METRICS

In this paper, we default to the information entropy $\mathcal{H}(\mathbf{z}_i)$ as a measure of its knowledge hardness,

$$\mathcal{H}(\mathbf{z}_i) = -\sum_j \sigma\big(\mathbf{z}_{i,j}/\tau\big)\log\big(\sigma\left(\mathbf{z}_{i,j}/\tau\right)\big). \tag{A.1}$$

To further evaluate the effects of knowledge hardness metrics, we consider another more complicated knowledge hardness metric proposed by KRD (Wu et al., 2023b), namely *Invariant Entropy*,

Table A4: Performance comparison of KRD and HGMD-mixup with two knowledge hardness metrics (Information Entropy and Invariant Entropy) under the transductive setting on seven datasets.

| Method | Knowledge Hardness | Cora | Citeseer | Pubmed | Photo | CS | Physics | ogbn-arxiv |
|---|---|---|---|---|---|---|---|---|
| MLPs | - | $59.58_{\pm 0.97}$ | $60.32_{\pm 0.61}$ | $73.40_{\pm 0.68}$ | $78.65_{\pm 1.68}$ | $87.82_{\pm 0.64}$ | $88.81_{\pm 1.08}$ | $54.63_{\pm 0.84}$ |
| GCNs | - | $81.70_{\pm 0.96}$ | $71.64_{\pm 0.34}$ | $79.48_{\pm 0.21}$ | $90.63_{\pm 1.53}$ | $90.00_{\pm 0.58}$ | $92.45_{\pm 0.53}$ | $71.20_{\pm 0.17}$ |
| KRD | Information Entropy | $83.87_{\pm 0.51}$ | $74.12_{\pm 0.47}$ | $81.24_{\pm 0.31}$ | $91.75_{\pm 1.46}$ | $\underline{94.21}_{\pm 0.37}$ | $93.90_{\pm 0.54}$ | $70.51_{\pm 0.24}$ |
| HGMD-mixup | Information Entropy | $\underline{84.66}_{\pm 0.47}$ | $\underline{74.62}_{\pm 0.40}$ | $\underline{82.02}_{\pm 0.45}$ | $\underline{93.33}_{\pm 1.31}$ | $94.16_{\pm 0.32}$ | $94.27_{\pm 0.63}$ | $\underline{71.09}_{\pm 0.21}$ |
| KRD | Invariant Entropy | $84.42_{\pm 0.57}$ | $\mathbf{74.86}_{\pm 0.58}$ | $81.98_{\pm 0.41}$ | $92.21_{\pm 1.44}$ | $94.08_{\pm 0.34}$ | $\underline{94.30}_{\pm 0.46}$ | $70.92_{\pm 0.21}$ |
| HGMD-mixup | Invariant Entropy | $\mathbf{84.89}_{\pm 0.35}$ | $74.48_{\pm 0.41}$ | $\mathbf{82.30}_{\pm 0.28}$ | $\mathbf{93.57}_{\pm 1.17}$ | $\mathbf{94.47}_{\pm 0.49}$ | $\mathbf{94.54}_{\pm 0.55}$ | $\mathbf{71.48}_{\pm 0.27}$ |

that defines the knowledge hardness of a GNN knowledge sample (node) $v_i$ by measuring the invariance of its information entropy to noise perturbations, as follows

$$\rho_i = \frac{1}{\delta^2} \underset{\mathbf{X}' \sim \mathcal{N}(\mathbf{X}, \mathbf{\Sigma}(\delta))}{\mathbb{E}} \left[ \left\| \mathcal{H}(\mathbf{z}_i') - \mathcal{H}(\mathbf{z}_i) \right\|^2 \right], \text{where } \mathbf{Z}' = f_\theta^{\mathcal{T}}(\mathbf{A}, \mathbf{X}') \text{ and } \mathbf{Z} = f_\theta^{\mathcal{T}}(\mathbf{A}, \mathbf{X}) \quad \text{(A.2)}$$

From the experimental results in Table. A4, we can make two observations that (1) Even the simplest information entropy metric is sufficient to yield state-of-the-art results. However, HGMD is also applicable to other knowledge hardness metrics in addition to information entropy; finer and more complicated hardness metrics, such as Invariant Entropy, can lead to more performance gains, regardless of for KRD or HGMD-mixup. (2) We made a fair comparison between HGMD-mixup and KRD by using the same knowledge hardness metrics. Despite the fact that HGMD-mixup does not involve any additional parameters in the distillation, HGMD-mixup outperforms KRD in 12 out of 14 metrics across seven datasets. In short, the choice of knowledge hardness metrics is an open problem, and our HGMD is compatible with a wide range of different knowledge hardness metrics.

