# OpenReview forum: "HGMD: Rethinking Hard Sample Distillation for GNN-to-MLP Knowledge Distillation"
_ICLR.cc/2024/Conference — Submitted to ICLR 2024_

### Official Review · Reviewer_RA2Z · 2023-10-26

**Soundness:** 2 fair
**Presentation:** 3 good
**Contribution:** 2 fair
**Rating:** 5
**Confidence:** 4

**Summary:**

This paper proposes a GNN-to-MLP distillation framework named HGMD. The proposed HGMD formulates the hardness of knowledge samples as their information entropy, based on which HGMD assigns a hardness-aware subgraph to each node and then distills subgraph-level knowledge from GNNs to MLPs. Experiments demonstrate that HGMD outperforms most of competitors in terms of the accuracy of student models.

**Strengths:**

1. This paper proposes two hardness-aware distillation schemes, i.e., HGMD-weight and HGMD-mixup. They do not introduce additional learnable parameters but still outperform most of competitors.
2. The experiment in Fig. 1(c) brings new insights into the GNN-to-MLP distillation, that is, the entropy of the GNN knowledge of most nodes misclassified by student MLPs is higher than that of the knowledge of other nodes.

**Weaknesses:**

1. The authors may want to explain the motivation for distilling teacher GNNs that are not very large (e.g., GCN, GraphSAGE, and GAT with two or three layers).
2. The theoretical analysis of the phenomenon of “the larger the entropy of GNN knowledge, the more difficult it is for MLPs to learn” is missing.
3. The logical connection between Fig. 1(b) and the main claim of this paper, i.e., “the hardness of the GNN knowledge is important to the GNN-to-MLP distillation” is unclear.
4. Some concerns about the inductive setting are as follows.

    - The authors may want to detail the experiments in the inductive setting, e.g., providing the proportion of testing nodes that are invisible to models during training. The missing of the details make the results in the inductive setting unconvincing.

    - I suggest the authors focus on the inductive setting rather than the transductive setting, where student MLPs can directly fit their teachers’ outputs on testing nodes, as they are available to the students during training. This means that it is trivial for the students to achieve comparable performance to their teachers in the transductive setting.

**Questions:**

See Weaknesses.

---

### Official Review · Reviewer_X3Yr · 2023-10-28

**Soundness:** 2 fair
**Presentation:** 2 fair
**Contribution:** 1 poor
**Rating:** 5
**Confidence:** 5

**Summary:**

This paper focuses on GNN-to-MLP knowledge distillation, and revisits the knowledge samples (nodes) in teacher GNNs from the perspective of hardness in addition to correctness, and identifies that hard sample distillation may be a major performance bottleneck of existing distillation algorithms. To tackle this problem, the authors propose a non-parametric HGMD framework, which models the hardness of different GNN knowledge samples based on information entropy and then extracts a hardness-aware subgraph for each sample separately. Finally, two hardness-aware distillation schemes (i.e., HGMD-weight and HGMD-mixup) are devised to enhance the framework.

**Strengths:**

1. The motivation of this paper is clear and the presentation is easy to follow.

2. The idea of combining a hardness-based Loss Weighting scheme with subgraph-level supervision and mixup is somewhat reasonable, and performs well in some graph datasets.

**Weaknesses:**

1. The proposed method comprises some widely-used tricks, such as a hardness-based Loss Weighting[1], subgraph-level distillation[2][3], and mixup[4]. Therefore, the novelty is limited.

2. The listed results of GLNN[5] and NOSMOG[3] are completely inconsistent with those of its original paper, especially the results in large-scale datasets ogbn-arxiv and products.

3. The performance improvement of the proposed method is not significant in large-scale datasets, which is important in GNN-to-MLP distillation.

[1] Helong Zhou, Liangchen Song, Jiajie Chen, Ye Zhou, Guoli Wang, Junsong Yuan, and Qian Zhang. Rethinking soft labels for knowledge distillation: A bias-variance tradeoff perspective. arXiv preprint arXiv:2102.00650, 2021.

[2] Lirong Wu, Haitao Lin, Yufei Huang, Tianyu Fan, and Stan Z Li. Extracting low-/high- frequency knowledge from graph neural networks and injecting it into mlps: An effective gnn-to-mlp distillation framework. In Proceedings of the AAAI Conference on Artificial Intelligence, 2023.

[3] Tian Y, Zhang C, Guo Z, et al. Learning mlps on graphs: A unified view of effectiveness, robustness, and efficiency[C]//The Eleventh International Conference on Learning Representations. 2023.

[4] Sami Abu-El-Haija, Bryan Perozzi, Amol Kapoor, Nazanin Alipourfard, Kristina Lerman, Hrayr Harutyunyan, Greg Ver Steeg, and Aram Galstyan. Mixhop: Higher-order graph convolutional architectures via sparsified neighborhood mixing. In international conference on machine learning, pages 21–29. PMLR, 2019.

[5] Zhang S, Liu Y, Sun Y, et al. Graph-less Neural Networks: Teaching Old MLPs New Tricks Via Distillation[C]//International Conference on Learning Representations. 2021.

**Questions:**

1. Why are the listed results of GLNN and NOSMOG completely inconsistent with those of its original paper?
2. Why is the performance improvement of the proposed method not significant in large-scale datasets?

---

### Official Review · Reviewer_61RB · 2023-11-02

**Soundness:** 2 fair
**Presentation:** 2 fair
**Contribution:** 2 fair
**Rating:** 5
**Confidence:** 4

**Summary:**

This paper proposes a GNN-to-MLP distillation method, namely HGMD, which distills subgraph-level knowledge from teacher GNNs into the corresponding nodes of student MLPs. Experiments demonstrate that HGMD outperforms the state-of-the-art competitors on several small datasets.

**Strengths:**

1. The idea of hard sample distillation is interesting.
2. Experiments demonstrate that HGMD outperforms the state-of-the-art competitors on several small datasets.

**Weaknesses:**

1. The accuracy rankings in Figure 1.(a) are difficult to support the claim that distillation from a larger teacher with more parameters and high accuracy may be inferior to distillation from a smaller and less accurate teacher, as the used baselines share similar performance. For example, the authors report GCN outperforms GAT on the Cora dataset, while [1] reports GAT outperforms GCN on the Cora dataset. This is because their average accuracy is similar while the corresponding standard deviation is large (the avearages+-standard deviations of GCN and GAT on the Cora dataset are 81.88+-0.75 and 82.80+-0.47 respectively in [1]). Some suggestions are as follows.
	1. The authors may want to evaluate a statistically significant difference. Moreover, in my opinion, the accuracy is more important than the accuracy rankings.
	2. The authors may want to compare the state-of-the-art GNNs from the top spots on the OGB leaderboards [2] with the vanilla GCN.
2. The authors claim that Pubmed is a small dataset and CS is a large dataset. However, the number of nodes in CS (# Nodes=18,333) is less than that in Pubmed (# Nodes=19,717) from Table A1. In my opinion, the small datasets include Cora, Citeseer, Pubmed, Photo, CS, and Physics.
3. The improvement on the large datasets (ogbn-arxiv and products) is marginal. The authors may want to evaluate a statistically significant difference against the second-best result.
4. The motivation for GNN-to-MLP distillation is unclear. In Section 4, the authors mainly focus on the transductive setting, whose metric is accuracy rather than inference time. Therefore, GNN-to-GNN distillation is more promising than GNN-to-MLP distillation in this setting, as MLP suffers from suboptimal prediction performance according to the claim in Section 1.


[1] GNNAutoScale: Scalable and Expressive Graph Neural Networks via Historical Embeddings. ICML 2021.

[2] https://ogb.stanford.edu/docs/leader_nodeprop/.

**Questions:**

See Weaknesses.

---

### Official Review · Reviewer_bK6B · 2023-11-10

**Soundness:** 2 fair
**Presentation:** 3 good
**Contribution:** 3 good
**Rating:** 5
**Confidence:** 4

**Summary:**

This paper revisits the knowledge samples in teacher GNNs from the perspective of hardness rather than correctness and proposes two hardness-aware distillation schemes (i.e., HGMD-weight and HGMD-mixup) to distill subgraph-level knowledge from teacher GNNs into the corresponding nodes of student MLPs.

**Strengths:**

1. It is good to evaluate several SOTA KD methods on eight datasets.
2. It is good to rethink that “better teacher, better student” does not always hold true.

**Weaknesses:**

1. Authors use “hard sample” many times, but its definition is missing. This can be confusing, especially in Section 2, where authors claim that most of the existing GNN-to-MLP KD methods made little effort on hard samples. This argument is not convincing and lacks some evidence.
2. Implementation details are missing.
3. In evaluation (Section 4.2), it is better to provide some insights rather than only listing numbers.

**Questions:**

1. How to set $\beta$ in Eq. 8? How does it affect the performance? Can some ablation study be provided? Similar for $\alpha$ in Eq. 6.
2. In Table 1, seven datasets are evaluated, excluding one dataset. Is there a reason for that? Additionally, the comparison with GLNN appears to be particularly extensive—what is the rationale behind this focus?

---

### Meta-Review · Area_Chair_o51X · 2023-12-06

**Metareview:**

The paper considers the scenario of GNN to MLP knowledge distillation and specifically investigate the role of "hardness" rather than correctness during the distillation process. Given this perspective, two schemes are proposed.

The reviewers have found the high level idea interesting and -at least for small datasets - the experiments are mostly convincing.

However, all reviewers remain overall unconvinced about the approach. Firstly, they find that crucial details are missing from the presentation, such as definition of "hard samples" or the theoretical analysis of the entropy-related phenomenon referred to by reviewer RA2XZ. Besides, experimental results for larger datatasets is not as good

**Justification For Why Not Higher Score:**

The authors have not provided a rebuttal.

**Justification For Why Not Lower Score:**

N/A

---

### Decision · Program_Chairs · 2024-01-16

Reject